# Regular Exercise with Suicide Ideation, Suicide Plan and Suicide Attempt in University Students: Data from the Health Minds Survey 2018–2019

**DOI:** 10.3390/ijerph19148856

**Published:** 2022-07-21

**Authors:** Ke Ning, Chun Yan, Yanjie Zhang, Sitong Chen

**Affiliations:** 1School of Physical Education and Sport, Shaanxi Normal University, Xi’an 710119, China; ningke@snnu.edu.cn; 2School of Economics and Management, Shannxi Xueqian Normal University, Xi’an 710061, China; 3Physical Education Unit, School of Humanities and Social Science, The Chinese University of Hong Kong, Shenzhen 518172, China; elite_zhangyj@163.com; 4Institute for Health and Sport, Victoria University, Melbourne 8001, Australia; sitong.chen@live.vu.edu.au

**Keywords:** exercise, suicide, young adult, survey

## Abstract

Background: Participating in exercise has been frequently recognized as a protective factor of suicide-related outcome (e.g., suicidal ideation) in children and adolescents, albeit with less of a focus on university/college students (especially using nationally representative sample). This study aimed to explore the associations between regular exercise with suicide ideation, plan, and suicide attempts using the data from Health Minds Survey (2018–2019 round). Methods: Using the cross-sectional data from a nationally representative sample (n = 62,026; mean age = 23.1 years) with self-reported information on exercise and the three suicide-related measures, binary logistic regression was used to estimate the associations of exercise with suicide ideation, plans, and attempts, respectively, while controlling for age, gender, being an international student or not, and race/ethnicity. Results: Compared with university students reporting five or more hours for exercise a week, those reporting less were more likely to report *yes* in terms of suicide ideation and a similar association was also observed in exercise and suicide plan. However, exercise was not significantly associated with suicide attempts. Conclusion: Spending more time exercising may be a protective factor against suicide ideation and plan for university students. Owing to the cross-sectional design nature, our research findings should be further investigated for confirmation or negation.

## 1. Introduction

Nowadays, with the increase of social competitive pressure, an increasing number of population experience poor well-being, including behavioral issues, stress, and depression [1,2]. The presence of mental disorders can easily result in negative health outcomes, such as suicidal behavior [3]. Notably, death by suicide is a leading factor of global mortality. According to the existing study, it reported an estimated 11.4 suicides every 100,000 people, and there are 804,000 suicide deaths worldwide [4]. The corresponding finical burden of suicide was approximately 93.5 billion [5]. Nevertheless, available data have shown that the number of suicides continues to rise rapidly in young people due to various reasons (e.g., depression, poor peer relationship, family discord, and social contagion) [6,7]. In order to reduce suicide deaths, both researchers and practitioners call for an urgent in fighting against this public health issue. Within research on suicide-related outcomes, suicidal ideation, plans, and attempts are the three core measures.

Previous works reported that physical activity provided a benefit to psychological health. This view is supported by Mikkelsen [8] et al., who reported the important role of exercise in alleviating the symptoms of depression, anxiety, and stress in individuals with mood disorders. Similar results were observed in young and medical students [9,10]. Moreover, physical activity might be a protective factor against suicide ideation in different populations subgroups. For instance, in a recent study, a negative relationship was found between physical activity and suicide risk among United States high school students. Kim et al. reported that individuals with high level of physical activity had lower suicidal ideation than those with low level of physical activity [11]. Plfedderer et al., found that daily engaging in physical activity was significantly associated with lower risk of suicidal ideation in United States adolescents [12]. This is also supported by a systematic review finding that participating more physical activity contributes to reducing suicidal ideation in adolescents and adults (including older adults) [13]. Thus, physical activity has been considered as a positive intervention for preventing suicidal ideation.

In addition to suicidal ideation, there are two other suicidal measures, including plans and attempts. Across the literature, some studies also examined the association between physical activity and suicidal plan and attempt respectively. For example, one study found that physical activity was significantly and negatively associated with suicidal attempt in university students, presenting a dose-response association [9]. In older adults, no engaging in physical activity was associated with high risks of suicidal attempt [14]. In terms of suicidal plans, there is also evidence suggesting the preventive role of physical activity in different populations. A study consisting of adolescents aged 14–18 years suggested a negative association between physical activity and suicidal planning [15]. These findings are supported by a study that both male and female students who participated in regular physical activity weekly had a decreased risk of suicidal plan and attempt compared to their inactive partners [16]. This information confirms the protective effect of more participations in physical activity on reduced likelihood of suicide-related measures. Based on this evidence, it would be possible that different types of physical activity can reduce the risk of suicide-related outcomes. However, this research assumption needs more evidence to confirm in different populations owing to scanty relevant research exists.

However, to our knowledge, little is known about the association between physical activity and suicidal ideation in university students, as much evidence was based on people aged under 18 years [13,17]. This inhibits researchers’ understanding on protecting university students far away from early death owing to suicide-related outcomes. In addition to this, the previous evidence focused on the association between overall physical activity instead of its specific components and suicidal ideation. Indeed, as an important source, exercise is defined as purposeful and healthful form of physical activity, which aims to increase individual’s health status. In this line, further exploring the health benefits of exercise in different population subgroups is also needed. However, research investigating the association between exercise and suicide-related outcomes is even extremely rare, particularly in university students. If researchers can further verify the association between exercise and suicide-related outcomes, it is highly likely to prevent university students’ early death, which can in turn reduce the health burden. In addition to the above research, few studies examined the associations of exercise with suicidal ideation, plans, and attempts. To fill the gaps in the literature, using a large-scale survey data would be a possible approach to achieve this research aim.

Therefore, based on the above literature, our study aimed to explore the association between exercise and suicide-related measures in university students using the data from Health Minds Survey (2018–2019). The Health Minds Survey is a yearly-surveyed health surveillance in US university students, generating a nationally representative sample to achieve our study aims.

## 2. Methods

### 2.1. Study Design and Participants

An accessible data from the 2018–2019 academic year of the Healthy Minds Study (HMS) was analyzed in this cross-sectional study (https://healthymindsnetwork.org/, accessed on 18 March 2022). The HMS is an internet-based survey investigating mental health and related problems among undergraduate and graduate students from 79 universities. A random sampling strategy was used to select a sample of 4000 students aged 18 from each university invited by email, while all students were invited to participant in this survey at small universities. The response rate of participating students was 16%. The HMS was approved by the Health Sciences and Behavioral Sciences Institutional Review Board at University of Michigan (application number: REP00000042). More study design details on the HMS survey can be accessed at the webpage (https://healthymindsnetwork.org/hms/, accessed on 18 March 2022).

### 2.2. Exposure (Exercise)

Exercise was described by using a question “In the past 30 days, how many hours per week on average did you spend exercising? (Including any exercise of moderate or higher intensity, where “moderate intensity” would be roughly equivalent to brisk walking or bicycling)”. The answers included: “Less than 1 h”, “2–3 h”, “3–4 h” and “5 or more hours”.

### 2.3. Outcome (Suicide Ideation, Plan and Attempt)

Suicide behaviors were measured with three aspects: suicidal ideation, suicide plan, and suicide attempt. Suicide ideation was measured using the question: “In the past year, did you ever seriously think about attempting suicide?” with responses of *yes or no*. Suicide plan and suicide attempt were measured using the question “In the past year, did you make a plan for attempting suicide?” (with *yes* or *no* answers) and “In the past year, did you attempt suicide?” (with *yes* or *no* answers), respectively. These questions were standardized measure to assess suicide ideation in previous surveillance and monitoring systems and their associated published research [18,19,20]. For this regard, this measure could be recognized with acceptable psychometric performance.

### 2.4. Covariates

Covariates included for further analysis consisted of sociodemographic characteristics, including age, gender (male, female, transgender, and other), race/ethnicity (African American/Black, American Indian or Alaskan Native, Asian American/Asian, Hispanic/Latino/a, Native Hawaiian or Pacific Islander, Middle Eastern, Arab, or Arab American, White, other), and international students (*yes*/*no*). These variables were treated as controlling variables in the further statistical analysis.

### 2.5. Statistical Analysis

In this cross-sectional study, study respondents’ characteristics and suicide-related outcome prevalence were examined with descriptive statistical methods. The multivariable logistic regression model was employed to explore the association between exercise and suicide-related outcome, controlling for age, gender, race/ethnicity, and international students. When establishing the logistic regressions, 5 or more hours of exercise was set as reference group while reporting no of suicide ideation, plan and attempt were set as reference group in the separate model. All results were presented as odds ratio (OR) with 95% confidence intervals. In the statistical analysis, we retained missing data since the variables of interest had different quantity of missing cases. All statistical analyses were performed using the SPSS 26.0 (IBM, Armonk, NY, USA) taking weighted sample into consideration. Statistical significance was set as *p* < 0.05.

## 3. Results

Table 1 presents the sample characteristics. The final sample included our analysis was 62,026 participants (mean age, 23.1 years old). The proportion of male was 31.9%. Most respondents were the US domestic students (91.3%) and majority White (64.2%). 26.8% of sample reported less than 1 h of exercise per week, 24.7% reported 2–3 h of exercise per week, 15.1% reported 3–4 h of exercise per week, and 24.6% reported 5 or more hours of exercise per week. About 11.5% of the study participants reported suicidal ideation. In terms of suicide plan and attempted suicide, the prevalence of yes was 3.9% and 0.9% respectively. Additionally, results based on sampling weighted are also reported in Table 1.

The results for the association between the exercise and suicide ideation are presented in Table 2. In the adjusted model, participants with less than 1 h of exercise per week were more likely to have suicide ideation compared to those with 5 or more hours of exercise per week (OR = 1.62, 95% CI [1.52, 1.72]). Moreover, a similarly significant association was observed in suicide plan, suggesting that engaging in less than 1 h of exercise per week resulted in 1.26 times greater odds for suicide plan comparted to engaging in 5 or more hours of exercise per week.

The suicide ideation was negatively associated with exercise. As detailed in Table 2 students who reported less participation in exercise (less than 1 h) had high risk of suicide ideation, compared with those participating more in exercise (5 or more hours) (OR = 1.62, 95% CI: 1.51–1.74; *p* < 0.01). study participants reporting 2–3 h or 3–4 h of exercise were less likely to report yes of suicide ideation as compared with 5 or more hours of exercise. Similar to the finding for suicide ideation, the significant association was observed for exercise and suicide plan, with students who reported less participation in exercise (less than 1 h) having an approximately 1.24-fold increased odds of having a suicide plan compared with those participating more in exercise (5 or more hours). However, there was no association between exercise and suicide attempt.

Table 3 depicts the association between exercise with suicide ideation, plan and attempt in the sample by gender. Among male, high risks of suicide ideation were associated with less than 1 h exercise per week (OR = 1.70, 95% CI: 1.55–1.87; *p* < 0.01), 2–3 h exercise per week (OR = 1.38, 95% CI: 1.25–1.52; *p* < 0.01), and 3–4 h exercise per week (OR = 1.20, 95% CI: 1.07–1.35; *p* = 0.002). High risks of suicide attempt were associated with less than 1 h exercise per week (OR = 1.57, 95% CI: 1.16–2.13; *p* = 0.004), 2–3 h exercise per week (OR = 1.73, 95% CI: 1.24–2.41; *p* = 0.01), and 3–4 h exercise per week (OR = 2.18, 95% CI: 1.40–3.38; *p* = 0.001). A similarly significant result was observed in suicide ideation (OR = 1.54, 95% CI: 1.41–1.69; *p* < 0.01) of female who attempted in less than 1 h exercise per week. Additionally, there were significant higher risks of suicide plan of female in the less than 1 h exercise per week (OR = 1.39, 95% CI: 1.16–1.67; *p* < 0.01), 2–3 h exercise per week (OR = 1.24, 95% CI: 1.02–1.50; *p* = 0.029), and 3–4 h exercise per week (OR = 1.49, 95% CI: 1.19–1.87; *p* = 0.01), compared to 5 or more hours of exercise per week. Other information on the association between exercise and suicide variables was presented in Table 3.

## 4. Discussion

The current cross-sectional study aimed to investigate the association between exercise and suicide ideation in university students using the data from Health Minds Survey (2018–2019). The findings also suggest that participating in more exercise (five or more hours per week) is negatively associated with suicide behaviors (ideation, plan) in university students rather than suicide attempt compared to those participated in regular exercise less than 1 h. Furthermore, compared to five or more hours of exercise per week, less time for exercise was significantly associated with high risk of suicide ideation in both male and female, suicide attempt in male and suicide plan in female. These findings may be helpful to design and implement better prevention for suicide deaths in university students, especially in the US. More detail discussions are presented below.

Our results support previous findings that students engaging in exercise less than one hour per week had high risk of suicide ideation and suicide plan compared to those engaging in exercise for five or more hours. With regarding to suicidal ideation, in 2009, a study found that lower risk of suicidal ideation was observed in university students who engaged in more exercise [16]. Moreover, in a recent population-based study investigating the suicidal ideation in Eswatini adults aged 18–60 years old, Mosta et al. reported that adults who did not participated in exercise had high risk of suicidal ideation [21]. With regarding to suicidal plan, an earlier study investigated the risk of suicidality among high school students, the result indicated a negative relationship between exercise participation and suicide plan [22]. Another study conducted by Li et al. reported a reverse link between exercise and suicide plan in American high school students. The results showed that students who never did exercise had approximately 1.31 greater times risk of suicide plan compared to those more frequently engaged in exercise [23]. Although there was not enough evidence on answering the mechanism of association between exercise and suicide-related behaviors, several potential explanations may exist. First, students could improve their self-esteem and sense of control after engaging in regular exercise, and thereby the symptoms of depression and suicidal behaviors might disappear. Second, biological mechanisms might be involved in the process of exercise, anti-depression, and suicidal behavior [24,25]. As such, exercise contributes to decreasing in the level of cortisol and increasing the brain neurotransmitters (i.e., BDNF, serotonin, dopamine), these positive hormones then play a key role in reducing the depression and suicide-related outcomes [25,26,27]. Third, students indeed could get acquainted with more peer partners to increase their communication opportunities, and then reducing the likelihood of negative thoughts (e.g., suicidal ideation, anxiety) [28,29]. On the contrary, students who do not take part in exercising may lose protective social networks and connections with friends and parents that contribute to improve healthy development and decline suicidal ideation and plan [30]. Collectively, our research findings suggested that the associations between exercise and suicide ideation and suicide plan are observed among US university students.

However, our research findings did not observe an association between exercise and suicide attempts in university students. This result is inconsistent with previous studies. For example, Vancampfort et al. reported a negative association between suicidal ideation in adolescents, suggesting that meeting physical activity guidelines is an effective protector for suicide plan [13]. Southerland et al. found adolescents who engaged in sport team demonstrated a lower risk of suicide-related behaviors (e.g., suicidal ideation, suicide attempt) [31]. Furthermore, one recent study suggested that exercise was associated with low risk of suicide attempt in university students in Norway [9]. The dissimilar findings may be associated with the different study methods. A previous study used several items to assess the suicidal attempt, whereas one question was used to evaluate in our study [9]. Another possible explanation is that the type of exercise or physical activity may influence the suicide attempt [32]. The students who participated in sports teams can obtain support from friends and coaches and can regulate their mental health, decreasing the risk of suicide attempt when they occur difficult time. This suggests that exercise with social support may be better in playing a role for suicide attempt prevention. Since there is no clear description of the type of exercise in the current study, it is possible to answer why in our study there was no significant association between exercise and suicide attempt in university students. Thus, future research can explore whether different types of exercise affect the relationship between exercise and suicide attempt. Overall, the strategy including exercise participation should be encouraged to prevent suicide behaviors in adolescents and university students.

In addition, with regard to the role of sex in the association between exercise and suicidal behaviors, the findings of the current study showed that less time for exercise was associated with high risk of suicidal ideation in both male and female. These findings were consistent with a previous study [33], which indicated that less time for exercise increases risk for suicide ideation in male and female adolescents. Moreover, our findings showed sex differences between suicide attempt in male and suicide plans in female. Less time spent exercising was associated with a higher risk of suicide attempts in male and suicide plans in female. Although the detailed explanation among males and females is unclear, the following hypothesis may be considered. The link between exercise and suicide attempt complicated in females than in males, such as body image dissatisfaction, may play an important role. Evidence suggested that females are more concerned about body image than males [34]. Females who are dissatisfied with their body image may not be able to reduce suicide plans even if they do more exercise [34]. Furthermore, it is likely that the ethnic distribution of students might be a factor, such as the association between exercise and suicide attempts was significantly greater among non-Hispanic white male than among students of other races [15]. Thus, more research is necessary to understand the role of sex in the association between exercise and suicidal behaviors.

### 4.1. Practical Implications

Based on the research findings, our study can provide some practical implications. The university health counselling center should develop multiple interventions for preventing suicidal ideation from the first year of university. It is a crucial time to cultivate students’ correct lifestyle when they are transitioning from youth to adults. In particular, doing more exercise is needed to prevent university students’ suicide-related behaviors, which can in turn reduce the rates of early death. As our measure of exercise did not specify the types, it is encouraged that individuals advocate university students to engage in various kinds of exercise, which can also lead to addition health benefits.

### 4.2. Study Strength and Limitation

Given that most studies have examined the association between exercise or physical activity and suicidal behaviors among people aged under 18 years old, as this population received more research attention, which can conclude that sufficient physical activity can prevent suicide-related outcomes. Our research findings extend the current literature and provide evidence that the associations between exercise and suicidal ideation and suicide plans are hardly varied across different population subgroups.

We have to admit some study limitations. First, we used self-reported measures to assess exercise and suicide ideation as well as other variables, which is subject to respondents’ recall bias and desirability. For example, the exercise measure was reported by students recall, and students might be difficult to answer correctly as a result of participating in different exercise regimes. These factors would negatively influence measurement validity and then might negatively affect our research findings. Second, the measurement is an inherent limitation. As our study was based on public data from a large-scale surveillance project, the measurements of this surveillance are unchangeable. To reduce the bias, with regard to exercise measurement, exercise time per week was classified based on recommendations from a previous study [35]. Third, low response rate (<20%) is a study limitation, and the sample was from the US university students, hence the findings may not be generalizable for university students in other countries. Fourth, owing to the Health Minds Survey is a cross-sectional survey and the nature of this design roles out causality inference. Fifth, we should admit that in terms of outcomes of suicide plan and attempt, the missing cases were extremely large. However, if we examined the analytical sample including measures of suicide plan and attempt, the sample size was also sufficient to reach a reliable result. Collectively, for the HMS data, it is necessary to increase sample size of cases of variable of interests. This would be beneficial to further increase the generalizability of the research findings. These inherent study limitations uncover our study strengths, including large sample size and its associated generalizability to university students in the US. It is expected that future studies can improve study based on the present study limitations.

## 5. Conclusions

Our study offers evidence on role of exercise against suicide-related outcomes in university students, suggesting the more participations in exercise may prevent suicide ideation, attempt and plan. This adds to evidence that to prevent early suicide death in university students, encouraging exercise is a good and feasible approach.

## Figures and Tables

**Table 1 ijerph-19-08856-t001:** Sample characteristics of this study.

		n	Mean/%	Weighted
Total		62,026		78,082
Age				
	Valid	62,025	23.1	23.2
	Missing	1		
Gender				
	Male	19,758	31.9	43.5
	Female	40,900	65.9	54.0
	Trans gender	230	0.4	0.4
	Other gender identity	1045	1.7	1.9
	Missing	93	0.1	0.2
International students				
	No	56,657	91.3	92.4
	Yes	5082	8.2	7.0
	Missing	287	0.5	0.6
Race/ethnicity				
	African American/Black	3535	5.7	8.0
	American Indian or Alaskan Native	165	0.3	0.3
	Asian American/Asian	7313	11.8	9.5
	Hispanic/Latino	4024	6.5	6.9
	Native Hawaiian or Pacific Islander	65	0.1	0.1
	Middle Eastern, Arab, or Arab American	892	1.4	1.4
	White	39,828	64.2	63.2
	Other/Self-identify	767	1.2	1.5
	Missing	5437	8.8	9.0
Exercise				
	Less than 1 h	16,650	26.8	27.6
	2–3 h	15,318	24.7	23.6
	3–4 h	9354	15.1	14.1
	5 or more hours	15,245	24.6	24.6
	Missing	5459	8.8	10.1
12-month Suicide Ideation				
	No	49,815	80.3	77.8
	Yes	7154	11.5	12.8
	Missing	5057	8.2	9.4
12-month Suicide Plan				
	No	3308	5.3	6.2
	Yes	2439	3.9	4.7
	Missing	56,279	90.7	89.1
12-month Suicide Attempt				
	No	5168	8.3	9.7
	Yes	530	0.9	1.1
	Missing	56,328	90.8	89.2

**Table 2 ijerph-19-08856-t002:** Association between exercise with suicide ideation and plan in the overall sample.

	*p*	OR	95%CI	
Suicide Ideation				
Less than 1 h	0.00	1.62	1.51	1.74
2–3 h	0.00	1.24	1.15	1.34
3–4 h	0.02	1.12	1.02	1.22
5 or more hours	REF			
Suicide Plan				
Less than 1 h	0.01	1.24	1.06	1.44
2–3 h	0.13	1.14	0.96	1.34
3–4 h	0.01	1.31	1.08	1.58
5 or more hours	REF			
Suicide Attempt				
Less than 1 h	0.26	0.86	0.67	1.12
2–3 h	0.08	0.78	0.59	1.03
3–4 h	0.30	0.84	0.61	1.17
5 or more hours	REF			

OR, odds ratio; CI, confidence interval; REF, reference group.

**Table 3 ijerph-19-08856-t003:** Association between exercise with suicide ideation and plan in the sample by gender (triple category).

	Male				Female				Other			
	*p*	OR	95%CI		*p*	OR	95%CI		*p*	OR	95%CI	
**Suicide ideation**												
Less than 1 h	0	1.7	1.55	1.87	0	1.54	1.41	1.69	0.323	1.19	0.85	1.66
2–3 h	0	1.38	1.25	1.52	0	1.37	1.25	1.5	0.493	1.14	0.79	1.63
3–4 h	0.002	1.2	1.07	1.35	0.025	1.13	1.02	1.26	0.2	0.75	0.48	1.17
5 or more hours	REF											
**Suicide Plan**												
Less than 1 h	0.174	1.14	0.94	1.38	0	1.39	1.16	1.67	0.739	1.11	0.6	2.04
2–3 h	0.252	0.89	0.72	1.09	0.029	1.24	1.02	1.5	0.658	0.87	0.46	1.64
3–4 h	0.294	0.88	0.69	1.12	0.001	1.49	1.19	1.87	0.116	0.53	0.24	1.17
5 or more hours	REF											
**Suicide Attempt**												
Less than 1 h	0.004	1.57	1.16	2.13	0.082	1.3	0.97	1.75	0.004	3.03	1.42	6.46
2–3 h	0.001	1.73	1.24	2.41	0.304	1.17	0.87	1.59	0.004	3.54	1.51	8.31
3–4 h	0.001	2.18	1.4	3.38	0.248	1.25	0.86	1.81	0.512	1.38	0.53	3.62
5 or more hours	REF											

OR, odds ratio; CI, confidence interval; REF, reference group.

## Data Availability

The datasets can be obtained from the website: https://healthymindsnetwork.org/ (accessed on 18 March 2022).

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
