# Peer review of "Regular Exercise with Suicide Ideation, Suicide Plan and Suicide Attempt in University Students: Data from the Health Minds Survey 2018–2019"

_ijerph, 2022, doi:10.3390/ijerph19148856_

Round 1

Reviewer 1 Report

Thank you very much for the opportunity to review the manuscript entitled "Regular Exercise with suicide ideation, suicide plan and suicide attempt in the US University Students: Data from the Health Minds Survey 2018-2019", which was sent to IJERPH.

This research fits into the main topic of the journal.

Despite the fact that this topic is interesting, and I really think that the authors have devoted a lot of effort to their research, this current manuscript still needs improvement.

First, the review needs improvement. The authors need to consider in more detail the psychological aspect of physical exercises, at least the elementary connection between the organizational processes and the mental state. It is also necessary to consider the medical aspect of suicide (for example, the journal of Suicidology"). In addition, recent research in the field of subjective well-being also indicates a link between exercise and the experience of well-being. This aspect has not been considered by the authors.

Secondly, the discussion also needs more thorough elaboration. The authors retell the obtained connections, cite as an example other studies where this connection is also revealed, but do not explain it psychologically and psycho-physiologically. It turns out to be a purely sociological study. The reader is interested to find out what the psychological/psychophysiological mechanism of this connection is.

Finally, the physical sex must also be taken into account, since the prevalence of suicide among males is higher than among women.

This manuscript is of real value to the readers of the International Journal of Environmental Research and Public Health.

I will be glad to review the revised manuscript.

Author Response

Reviewer 1

Thank you very much for the opportunity to review the manuscript entitled "Regular Exercise with suicide ideation, suicide plan and suicide attempt in the US University Students: Data from the Health Minds Survey 2018-2019", which was sent to IJERPH.

This research fits into the main topic of the journal. Despite the fact that this topic is interesting, and I really think that the authors have devoted a lot of effort to their research, this current manuscript still needs improvement.

Response: Thank you for your comments. The below is our response.

First, the review needs improvement. The authors need to consider in more detail the psychological aspect of physical exercises, at least the elementary connection between the organizational processes and the mental state. It is also necessary to consider the medical aspect of suicide (for example, the journal of Suicidology"). In addition, recent research in the field of subjective well-being also indicates a link between exercise and the experience of well-being. This aspect has not been considered by the authors.

Response: Thank you for your comments. we have revised our manuscript accordingly. Please see line 43 – 49 in the revised manuscript.

Secondly, the discussion also needs more thorough elaboration. The authors retell the obtained connections, cite as an example other studies where this connection is also revealed, but do not explain it psychologically and psycho-physiologically. It turns out to be a purely sociological study. The reader is interested to find out what the psychological/psychophysiological mechanism of this connection is.

Response: Thank you for your comments. we have revised our manuscript accordingly. Please see line 222 – 228 in the revised manuscript.

Finally, the physical sex must also be taken into account, since the prevalence of suicide among males is higher than among women.

Response: Thank you very much! Based on your comments and those from the other reviewer, we have run the analysis by sex. Please see that in Line 179 – 193 and Line 258 – 274 in the revised manuscript. However, owing to the sample size for different racial groups are disproportional and some subgroup’s sample size was insufficient for stratified analysis, it is inappropriate to conduce a race-stratified analysis. May you understand our request and situation.

This manuscript is of real value to the readers of the International Journal of Environmental Research and Public Health. I will be glad to review the revised manuscript.

Response: Thank you for your encouragement. We think our revision can address your research concerns. Thank you for your time and effort again!

Reviewer 2 Report

This is a manuscript that addresses a relevant issue in relation to possible protective and risk factors for suicidal behavior.

There is a very large number of typographical errors throughout the document.

The introduction is pertinent. It is recommended to avoid the use of the expression suicide commit and its derivatives, it is better to express them as suicide deaths. The statement in lines 36 to 38 is inaccurate. Since the increases in suicide rates have occurred in young people, while in older people it has decreased. Turecki et al.'s own cited work. Realize it. It is requested to adjust the affirmation according to the precise data.

Regarding the method, it is observed to be adequate for the objective. Although it can be deepened by comparing gender and racial group in their relationship with exercise and suicidal behavior.

The results are presented correctly.

The discussion does not seem adequate based on the findings. The results of the study mention that people who do less than 3 to 4 hours of exercise have a higher risk of presenting suicidal ideation and suicidal plan. Most of the discussion centers around exercising being a protective factor when their data mentions that getting a little exercise is a risk factor. The discussion must be changed with this logic.

Consequently, the conclusions should also be changed following their findings.

Ethical considerations must be included.

The references do not have the design required by the journal, particularly in journal references.

Author Response

Reviewer 2

This is a manuscript that addresses a relevant issue in relation to possible protective and risk factors for suicidal behavior. There is a very large number of typographical errors throughout the document.

Response: Thank you for your encouragement. We have made a proofread to reduce the relevant errors. Please see the revised manuscript.

The introduction is pertinent. It is recommended to avoid the use of the expression suicide commit and its derivatives, it is better to express them as suicide deaths. The statement in lines 36 to 38 is inaccurate. Since the increases in suicide rates have occurred in young people, while in older people it has decreased. Turecki et al.'s own cited work. Realize it. It is requested to adjust the affirmation according to the precise data.

Response: Thank you for your comments. We have changed the inappropriate expression “suicide commit” to “suicide deaths”. Moreover, the corresponding inaccurate statement has revised accordingly.

Regarding the method, it is observed to be adequate for the objective. Although it can be deepened by comparing gender and racial group in their relationship with exercise and suicidal behavior. The results are presented correctly.

Response: Thank you for your encouragement. We have revised the results section according to your comments. Very insightful! Please see Line 179 – 193 in the revised manuscript. However, owing to the sample size for different racial groups are disproportional and some subgroup’s sample size was insufficient for stratified analysis, it is inappropriate to conduce a race-stratified analysis. Instead, we only performed the sex-specific analysis. May you understand our request and situation.

The discussion does not seem adequate based on the findings. The results of the study mention that people who do less than 3 to 4 hours of exercise have a higher risk of presenting suicidal ideation and suicidal plan. Most of the discussion centers around exercising being a protective factor when their data mentions that getting a little exercise is a risk factor. The discussion must be changed with this logic.

Response: Thank you for your comments. We have revised the description in discussion. Line 208 – 210: Our results support previous findings that higher levels of exercise could be a protective factor against students engaging in exercise less than 1 hours per week had high risk of suicide ideation and suicide plan compared to those engaging in exercise 5 or more hours.

Consequently, the conclusions should also be changed following their findings.

Response: Thank you for your comments. We have revised the conclusion according to your suggestion.

Ethical considerations must be included.

Response: Thank you for your comments. We have added the ethical evidence in the method. “The HMS was approved by the Health Sciences and Behavioural Sciences Institutional Review Board at University of Michigan (application number: REP00000042).”

The references do not have the design required by the journal, particularly in journal references.

Response: Thank you very much! We have amended the reference format.

Reviewer 3 Report

The manuscript "Regular Exercise with suicide ideation, suicide plan and suicide attempt in the US University Students: Data from the Health Minds Survey 2018-2019" is an important work on suicide risk among emerging adults. The strength of this study is a representative large sample, clear and rigorous reporting of results. However, measurement of the dependent and independent variables is a main source of my concern.

Exercise was assessed in a long time duration (30 days). Some people can exercise inconsistently each of four weeks of the month, so the answer for the question may be biased. Moreover, it is unclear why the response scale was categorized as “Less than 1 hour”, “2-3 hours”, “3-4 hours” and “5 or more hours”. Could you refer to any theory for that? The sentences "This question has been used in many survey studies" (page 3, line 110) is not supported by any on them in references. The WHO recommends at least 150–300 minutes of moderate-intensity aerobic physical activity; or at least 75–150 minutes of vigorous-intensity aerobic physical activity; or an equivalent combination of moderate- and vigorous-intensity activity throughout the week. And just last week is a good measure of current physical activity status. So the measurement used in the study is questionable. There are also questionnaires measuring PA more accurately, like IPAQ for example. So it is necessary to explain why this exercise measure was selected for the study purpose. In addition, this weakness should be extensively discussed in the limitation section.

The second problem is assessing suicidal risk in as long period like one year! The association between assessing PA during last month and suicidal behavior during last year makes no sense in my understanding of behavioral and cognitive processes. The recall bias is also a serious problem, for that type of harmful behavior. Coincidence between exercise and suicidal risk may be apparent. In fact, I don't know what exactly was associated with.

Unfortunately, these problems cannot be resolved in revision. Therefore, I cannot recommend this work for publication.

Author Response

Reviewer 3

The manuscript "Regular Exercise with suicide ideation, suicide plan and suicide attempt in the US University Students: Data from the Health Minds Survey 2018-2019" is an important work on suicide risk among emerging adults. The strength of this study is a representative large sample, clear and rigorous reporting of results. However, measurement of the dependent and independent variables is a main source of my concern.

Response: Thank you for your time and effort to review our manuscript. Regardless of the decision you made, we appreciate your comments to help us improve the studies. The below is our responses.

Exercise was assessed in a long-time duration (30 days). Some people can exercise inconsistently each of four weeks of the month, so the answer for the question may be biased. Moreover, it is unclear why the response scale was categorized as “Less than 1 hour”, “2-3 hours”, “3-4 hours” and “5 or more hours”. Could you refer to any theory for that? The sentences "This question has been used in many survey studies" (page 3, line 110) is not supported by any on them in references. The WHO recommends at least 150–300 minutes of moderate-intensity aerobic physical activity; or at least 75–150 minutes of vigorous-intensity aerobic physical activity; or an equivalent combination of moderate- and vigorous-intensity activity throughout the week. And just last week is a good measure of current physical activity status. So the measurement used in the study is questionable. There are also questionnaires measuring PA more accurately, like IPAQ for example. So, it is necessary to explain why this exercise measure was selected for the study purpose. In addition, this weakness should be extensively discussed in the limitation section.

Response: Thank you very much for your suggestive comments. We have revised the sentence in the revised manuscript, please see Line 296 – 300. The measurement is an inherent limitation. As our study was based on public data from a large-scale surveillance project, which means the measurements of this surveillance are unchangeable. To reduce the bias, with regard to exercise measurement, exercise time per week was classified based on recommendations from a previous study (https://jamanetwork.com/journals/jamainternalmedicine/fullarticle/485155; A Prospective Study of Recreational Physical Activity and Breast Cancer Risk).

The second problem is assessing suicidal risk in as long period like one year! The association between assessing PA during last month and suicidal behavior during last year makes no sense in my understanding of behavioral and cognitive processes. The recall bias is also a serious problem, for that type of harmful behavior. Coincidence between exercise and suicidal risk may be apparent. In fact, I don't know what exactly was associated with.

Unfortunately, these problems cannot be resolved in revision. Therefore, I cannot recommend this work for publication.

Response: Thank you very much for your suggestive comments. As we used the public data, this means the measure are unchangeable; so, we retain the original measurement section. May you understand our request and situation.

Round 2

Reviewer 3 Report

It is impossible to assess the influence of recently assessed physical activity (as a predictor) on the prediction of suicidal behavior a year, half a year, or even a month earlier. For logical reasons, the predictor (measurement of physical activity) must precede the event it is intended to explain (suicidal ideation or suicidal behavior) and not follow chronologically, as it contradicts the causal relationship.

The only chance to save this article is to leave the analyses at the level of chi-square tests, i.e. associations. It just doesn't make sense to create a logistic regression model in the predictive relationship between these variables. From this point of view, it would be more reasonable to assume that suicidal thoughts from a few months ago are a predictor of physical activity in the last week. Such logic, however, seems just as absurd as the model proposed currently by the authors. 

In summary, I suggest removing the logistic analysis from the manuscript, with table 2 and table 3. There is no other possibility to publish the result of this study. As such, the abstract, results, discussion, and conclusions sections should be adequately changed to the removing parts.

Author Response

Dear reviewer,

Thank you very much for your time and suggestions.

However, if we revise the manuscript based on your comments, the research question cannot be addressed by this study. We know your comments are very constructive but they may not be helpful to our manuscript revision. So, you can make your decision and we are fine with the editors' final decision. We appreciate your effort and time to improve our manuscript.

Thank you very much

Kinds